# Approaches for combining primary care electronic health record data from multiple sources: a systematic review of observational studies

Daniel Dedman ![ORCID],[1,2] Melissa Cabecinha ![ORCID],[3] Rachael Williams,[1] Stephen J W Evans,[4] Krishnan Bhaskaran,[2] Ian J Douglas[2]

¹Clinical Practice Research Datalink, Medicines and Healthcare Products Regulatory Agency, London, UK
²Department of Non-communicable Disease Epidemiology, London School of Hygiene and Tropical Medicine, London, UK
³Research Department of Primary Care and Population Health, University College London, London, UK
⁴Department of Medical Statistics, London School of Hygiene and Tropical Medicine, London, UK

**Correspondence to**
Daniel Dedman;
Daniel.Dedman@mhra.gov.uk

## ABSTRACT

**Objective** To identify observational studies which used data from more than one primary care electronic health record (EHR) database, and summarise key characteristics including: objective and rationale for using multiple data sources; methods used to manage, analyse and (where applicable) combine data; and approaches used to assess and report heterogeneity between data sources.

**Design** A systematic review of published studies.

**Data sources** Pubmed and Embase databases were searched using list of named primary care EHR databases; supplementary hand searches of reference list of studies were retained after initial screening.

**Study selection** Observational studies published between January 2000 and May 2018 were selected, which included at least two different primary care EHR databases.

**Results** 6054 studies were identified from database and hand searches, and 109 were included in the final review, the majority published between 2014 and 2018. Included studies used 38 different primary care EHR data sources. Forty-seven studies (44%) were descriptive or methodological. Of 62 analytical studies, 22 (36%) presented separate results from each database, with no attempt to combine them; 29 (48%) combined individual patient data in a one-stage meta-analysis and 21 (34%) combined estimates from each database using two-stage meta-analysis. Discussion and exploration of heterogeneity was inconsistent across studies.

**Conclusions** Comparing patterns and trends in different populations, or in different primary care EHR databases from the same populations, is important and a common objective for multi-database studies. When combining results from several databases using meta-analysis, provision of separate results from each database is helpful for interpretation. We found that these were often missing, particularly for studies using one-stage approaches, which also often lacked details of any statistical adjustment for heterogeneity and/or clustering. For two-stage meta-analysis, a clear rationale should be provided for choice of fixed effect and/or random effects or other models.

### Strengths and limitations of this study

► Our systematic review identified the increasing number of published observational studies, which specifically used primary care electronic health record (EHR) data from two or more sources.
► There were no restrictions on study design, exposures or outcomes.
► In the absence of relevant Medical Subject Heading terms, the search strategy relied on an extensive list of named primary care EHR databases to achieve as comprehensive coverage as possible.
► The selected publications were independently reviewed by two researchers.
► The findings of this review may not apply to multi-database studies, which did not use primary care EHR data sources.

## INTRODUCTION

Multi-database observational studies are increasingly common. They are conducted for two main reasons: to compare results across diverse populations and healthcare settings, or to combine the data to increase statistical power. Primary care electronic health record (EHR) databases are particularly valuable because they provide longitudinal data on individuals, often over many years, and typically contain richer information on a broader range of exposures, risk factors and health outcomes than administrative databases.[1–3] Although individual primary care EHR databases are often relatively small, covering a single region or other national population subset, their growing availability in recent years is likely to further increase the importance of non-interventional studies, which combine these databases. Guidelines identifying best practice in this context have yet to be established but would be of clear benefit for researchers working with multiple databases. An important preliminary step is to describe current practice, but there is no comprehensive summary of studies which

used two or more primary care EHR databases, and the methods for combining them.

One previous systematic review focused on multi-database pharmacoepidemiology studies with a pre-planned approach to combine data to evaluate drug-outcome associations.[4] In that review, studies were not limited according to the types of databases used, but descriptive studies and those which did not combine results from different databases were excluded. The authors found that for data management arrangements, analysis of heterogeneity and methods for combining data reporting were often inadequate, making interpretation of study results more challenging. Since the focus of that review was pharmacoepidemiology studies and a wide range of database types, a broader view of combined primary care EHR data for any study purpose remains lacking.

The aim of this systematic review was to identify and describe the full range of completed studies which brought together primary care EHR data from two or more sources. The specific objectives were to summarise key study characteristics, including the main reasons or motivations for including data from different EHR databases; to describe the methods used to manage, analyse and (where applicable) combine data; and describe the approaches used to assess and report heterogeneity between primary care EHR data sources.

## METHODS

The review considered all multi-database studies, published in English language between 2000 and 2018, and which included at least two different primary care EHR databases or data sources, irrespective of whether other types of database were also included. Primary care EHR data was defined as data collected by primary care clinicians and related staff for the purpose of diagnosis, treatment, management and delivery of care of individual patients, and could include information contributed by other care providers.[5] It excluded data generated primarily for administrative purposes such as health insurance claims data, where the motivation for recording is different. Primary care EHR databases were considered irrespective of whether they were 'vertically' linked (ie, linked at the individual patient level) to another data source such as a disease registry or dispensing database. Each 'vertically' linked primary care EHR database was treated as a single data source. Apart from the specific focus on primary care EHR databases, no other restrictions were applied in terms of populations, geography, study period, exposure, outcome or study design.

A previous systematic review highlighted the challenge of identifying multi-database studies, for which no specific Medical Subject Heading (MeSH) terms exist.[4] An alternative approach was, therefore, used, based on a comprehensive list of named primary care EHR databases compiled from two online registers[6 7] and one systematic review of primary care data

collection projects.[8] For each named database, a keyword search was generated and run on Medline, and the results combined. Abstracts of published studies identified in this search were scanned for additional terms and phrases, which might be used to describe the primary care EHR data sources, and from these additional keyword searches were generated. The final search strategy (see online supplemental material) was used to identify studies in Medline and Embase databases published between January 2000 and May 2018.

Titles and abstracts of all retrieved studies were screened for eligibility by one reviewer (DD). A random 20% sample was also screened by a second reviewer (MC) and showed very good agreement between the two. Reference lists of papers selected for full review were hand searched for additional studies.

Full text was obtained for all papers selected during the initial screening, and read by two reviewers (DD and MC), who independently completed the final eligibility assessment and data extraction. Each reviewer extracted standardised information from the study publication and online supplemental materials (where available), which was entered into the review database (Microsoft Access) via electronic data collection form developed by one of the reviewers (DD), and pilot tested with seven studies. Information extracted included data sources used, main objectives, study design, study populations, exposure, comparators and outcomes, data management arrangements and statistical methods. All discordant results were reviewed, and the final designation agreed by both reviewers. No additional information was sought from investigators.

Studies were classified as analytical if they estimated an exposure–outcome association, or descriptive otherwise. We noted whether and how between-database heterogeneity was assessed, how this informed the decision to combine the data and whether a one-stage meta-analysis of pooled individual patient data (IPD) or two-stage meta-analysis of study-specific effect estimates was used, as well as choice of fixed-effects (FE) versus random-effects (RE) models. A clear rationale for using multiple data sources was not always stated, but in some instances could be inferred. For analytical studies where results were combined using one-stage or two-stage meta-analysis, unless stated otherwise the rationale was assumed to be an increase in the statistical precision of the exposure–outcome effect estimate. Three main models for data management and analysis were considered, based on previous reviews[4 9–11]: a fully centralised model for management and analysis of the raw data provided by each contributing database; a fully distributed model where all data management and analysis was undertaken locally, and only fully aggregated results were shared; and a partially distributed model with local extraction and data management to generate standardised patient level or partially aggregated datasets

in standardised format, which were then shared for final centralised analysis. Partially aggregated (or semi-aggregated) data summarise information on more than one individual (thereby enhancing privacy protection) while still allowing the pooling of data across databases for further analysis, including one-stage meta-analysis. Examples include total person time and event counts for groups of patients sharing the same characteristics.

We noted whether studies used a global common data model (CDM) such as those from OMOP (Observational Medical Outcomes Partnership)[12 13] or Sentinel,[14 15] and whether they were part of a wider programme or initiative for developing database networks and methods for combining results—such as IMI-PROTECT (Innovative Medicines Initiative Pharmacoepidemiological Research on Outcomes of Therapeutics by a European Consortium),[16 17] EU-ADR(European Union Adverse Drug Reaction database network)[18] or ARITMO (Arrhythmogenic Potential of Medicines project).[19 20]

The focus of the review was on describing the range of multi-database studies and methods for combining primary care EHR data, rather than evaluating evidence of effectiveness of specific interventions. Given this, and in the absence of consensus or validated reporting guidelines for multi-database studies, no formal assessment of risk of bias or study quality was attempted.

The study protocol, including the final Medline search strategy and details of data items extracted, is provided in online supplemental file 1.

## Patient and public involvement
Patients or the public were not involved in the design, or conduct, or reporting, or dissemination plans of our research.

## RESULTS
The Medline and Embase searches returned 6049 results, and a further 5 were identified by hand searches. After initial screening of abstracts, 138 papers were selected for full text review, and 109 were included in the final review (figure 1). Summary information on the included studies is provided in online supplemental table S1.

Included studies used data from 38 different primary care EHR databases. Most studies (98, 89%) used 2 or 3 primary care EHR databases, and 43 studies (39%) also included 1 or more non-primary care EHR database. All but 3 studies used exclusively European primary care data, and 35 (32%) used data from a single country. Details of the primary care EHR databases used in included studies are summarised in online supplemental table S2.

The annual number of published studies increased over time (online supplemental figure S1), with fewer than 10 studies per year between 2003 and 2013, while a peak in 2016 (25 studies) included 10 studies published

in a special supplement on the IMI-PROTECT research programme.[16 17]

General characteristics of included studies are given in table 1. More than half (62 studies, 57%) were classified as analytical. Most studies (76, 70%) examined safety, effectiveness or utilisation of specific drugs accounted for, while 21% (23 studies) were disease epidemiology or risk prediction studies with no specific focus on pharmacological therapies.

Cohort studies were the most common study design (72 studies, 66%), and the majority of these were descriptive. Six studies included more than one study design.

The most common approach for data management and analysis was a fully centralised model (44 studies, 40%). A fully distributed model was used in 23 studies (21%), including 15 studies conducted as part of the IMI-PROTECT programme.[16 21] A partially distributed approach was used in 20 studies (18%), including 9 studies from the EU-ADR programme[18] and 5 from the ARITMO project.[19 20] No studies reported using a CDM.

Methodological aspects of the 62 analytical studies are summarised in table 2. All 23 case–control studies employed individual matching, and used conditional logistic regression to estimate adjusted ORs, but for cohort studies a range of statistical approaches was used.

In 22 analytical studies (35%), data were not combined, and all results were presented separately for each database—usually in order to describe and assess the consistency of findings in different populations or settings, using a common study protocol and analysis approach. In the remaining five studies, risk prediction models were developed in one primary care EHR database, and validated using a second primary care EHR database from the same country (the UK).[22–26]

In 40 analytical studies (65%), including the majority (20/23) of case–control studies, and half (18/34) of cohort studies, some form of pooled analysis or meta-analysis was undertaken. A one-stage meta-analysis of pooled IPD or partially aggregated data was undertaken in 29 studies (47%). In 19 of these studies, no assessment or discussion of between-database heterogeneity was provided, and only 4 studies reported any form of analytical adjustment for the clustered nature of the pooled data—in each case by including database as a covariate in a multiple regression model, with one study also including interaction terms between database and covariates.[27] Two-stage meta-analysis of database-specific effect estimates was used in 21 studies (34%), of which 14 presented some discussion or formal assessment of heterogeneity. The choice of FE or RE models was not clearly justified in most studies, though in four cases model choice was based on formal tests of heterogeneity (results not shown).

In 41 analytical studies (66%), separate effect estimates were reported for each database. Between database heterogeneity was formally assessed (most commonly using the $I^2$ statistic) in 17 studies (27%) and was discussed but not formally assessed in a further 6 studies (10%). In 17 studies (27%)—all of which used a one-stage approach—results

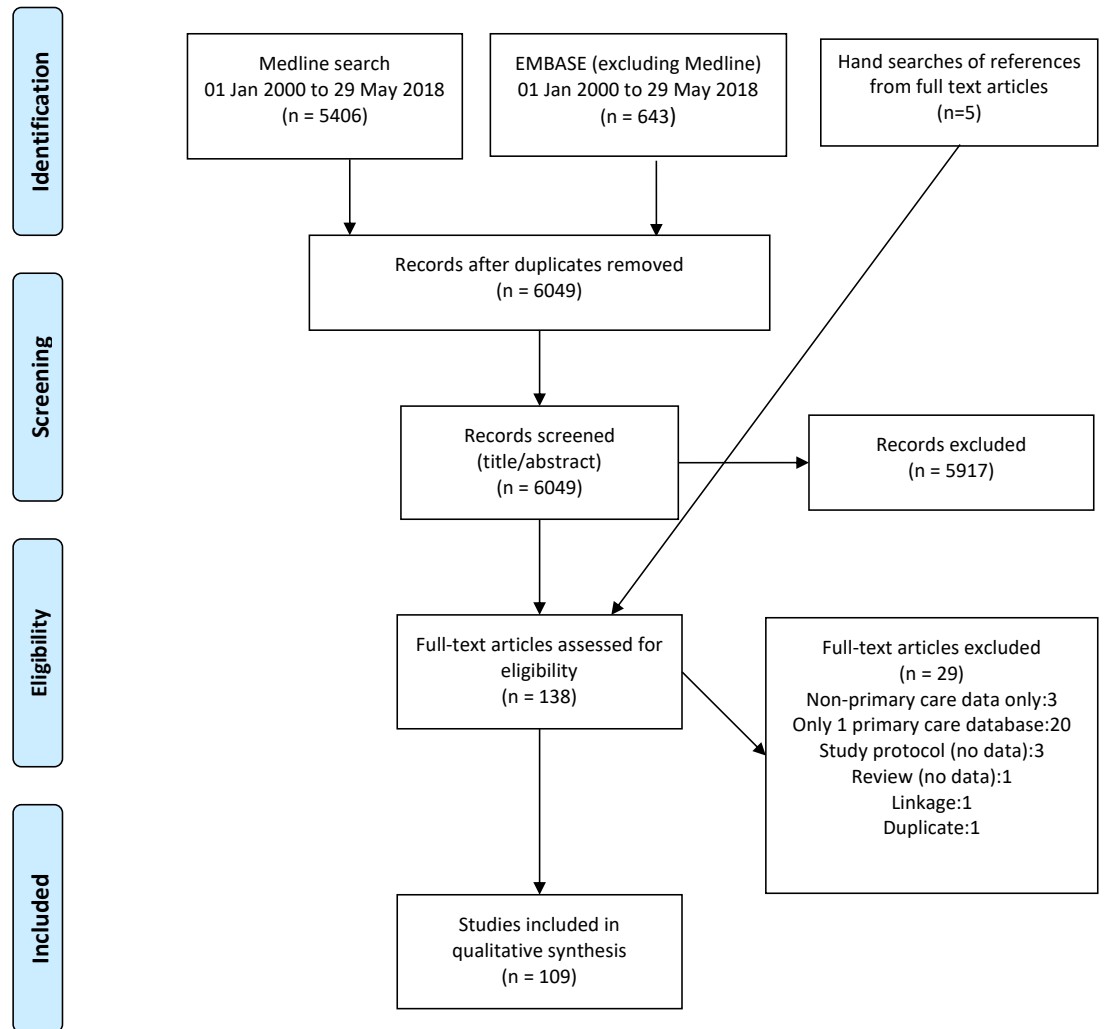

**Figure 1**  Selection and inclusion of studies for systematic review (adapted from Moher *et al*[70]).

were presented for the combined dataset only. Only 2 of these 17 studies additionally included a two-stage meta-analysis and reported heterogeneity statistics.

Ten studies reported comparable results (in the main paper or in supplementary materials) using two or more analytic approaches, such as both one-stage and two-stage meta-analysis (nine studies), or two-stage analysis with both FE and RE models (one study), summarised in figure 2. In most cases one-stage and two-stage meta-analysis gave very similar results. CIs from RE meta-analysis were in almost every case wider than the corresponding FE or one-stage model estimates. In one study (review database id: 117), point estimates from one-stage and two-stage meta-analyses differed appreciably, and RE model CIs were wider, partly because some databases with very small numbers of cases were excluded.[28] In one self-controlled case series study examining risk of upper gastrointestinal bleeding with different drugs alone and in combination, the point estimates from one-stage model (ignoring clustering) and two-stage RE effects models were very

different (review database id: 113a-d), and CIs from the two-stage models were much wider.[29]

## DISCUSSION

This systematic review identified 109 multi-database studies, which used data from 2 or more primary care EHR data sources, the great majority of which were from European countries. Just under half were descriptive studies undertaken either to compare patterns and trends in different populations or settings, or to assess comparability of different EHR databases from similar populations. In these descriptive studies, and in a third of analytical studies, there was no attempt to combine results from the different databases. Where data was combined, a one-stage meta-analysis of pooled IPD was used more often than two-stage meta-analysis of database-specific results. Reporting of statistical methods for one-stage analyses in particular was suboptimal: in all but four such

**Table 1** General characteristics of included studies, including objective, rationale and study design

| | Study type | | | | |
|---|---|---|---|---|---|
| | **Analytical** | **Descriptive** | **Other*** | All | % |
| **All Studies** | 62 | 45 | 2 | 109 | |
| Study objective | | | | | |
| Drug safety | 37 | 5 | | 42 | 38.5 |
| Drug utilisation | 3 | 24 | 1 | 28 | 25.7 |
| Disease epidemiology | 7 | 9 | | 16 | 14.7 |
| Disease risk prediction | 5 | 2 | | 7 | 6.4 |
| Drug comparative effectiveness | 6 | | | 6 | 5.5 |
| Methodology/data quality | 2 | 3 | 1 | 6 | 5.5 |
| Health services research | 2 | 2 | | 4 | 3.7 |
| Main rationale for using multiple data sources (stated or inferred) | | | | | |
| Describe trends and variation between countries or settings | 2 | 28 | | 30 | 27.5 |
| Increase study power | 24 | 2 | | 26 | 23.9 |
| Examine consistency of findings in different settings (using a standardised approach or common study protocol) | 19 | 3 | | 22 | 20.2 |
| Compare availability/quality of data in each source | 2 | 7 | 2 | 11 | 10.1 |
| Validation of findings in a second data source | 5 | 3 | | 8 | 7.3 |
| Not clearly stated | 10 | 2 | | 12 | 11.0 |
| Databases per study: primary care EHR only | | | | | |
| Mean | 2.4 | 2.9 | 2.5 | 2.6 | |
| Median (range) | 2 (2–4) | 3 (2–5) | 2.5 (2–3) | 2 (2–5) | |
| Databases per study: all types | | | | | |
| Mean | 3.1 | 4.3 | 5.0 | 3.7 | |
| Median (range) | 2 (2–8) | 4 (2–8) | 5 (2–8) | 3 (2–8) | |
| Database setting | | | | | |
| Single country | 25 | 9 | 1 | 35 | 31.8 |
| Multi-country | 37 | 36 | 1 | 75 | 68.2 |
| Study design† | | | | | |
| Cohort study | 33 | 38 | 1 | 72 | 66.1 |
| Case–control study | 23 | 0 | 0 | 23 | 21.1 |
| Cross-sectional | 1 | 6 | 1 | 8 | 7.3 |
| Self-controlled designs | 7 | 0 | 0 | 7 | 6.4 |
| Other | 0 | 1 | 1 | 2 | 1.8 |
| Interrupted time series | 1 | 0 | 0 | 1 | 0.9 |
| Data management and analysis model (stated or inferred) | | | | | |
| Centralised management and analysis: raw data shared | 32 | 11 | 1 | 44 | 40.4 |
| Distributed management and analysis: aggregated results shared | 11 | 12 | | 23 | 21.1 |
| Distributed management+centralised analysis: patient level or partially aggregated data shared | 11 | 8 | 1 | 20 | 18.3 |
| Not described | 8 | 14 | | 22 | 20.2 |
| Study drug (ATC chapter) | | | | | |
| Nervous system | 11 | 8 | | 19 | 17.4 |
| Respiratory system | 9 | 5 | | 14 | 12.8 |
| Musculoskeletal system | 9 | 3 | | 12 | 11.0 |
| Multiple categories | 7 | 4 | | 11 | 10.1 |

Continued

| | Study type | | | | |
|---|---|---|---|---|---|
| **Table 1** Continued | **Analytical** | **Descriptive** | **Other*** | **All** | **%** |
| Alimentary tract and metabolism | 6 | 3 | 1 | 10 | 9.2 |
| Antiinfectives for systemic use | 4 | 4 | | 8 | 7.3 |
| Cardiovascular system | 2 | 2 | | 4 | 3.7 |
| Genito urinary system and sex hormonesc | 2 | 1 | | 3 | 2.8 |
| Blood and blood forming organs | | 2 | | 2 | 1.8 |
| Dermatologicals | | 1 | | 1 | 0.9 |
| N/A | 12 | 12 | 1 | 25 | 22.9 |
| Study condition (ICD-10 chapter) | | | | | |
| Diseases of the circulatory system (I00–I99) | 15 | 4 | | 19 | 17.4 |
| Diseases of the respiratory system (J00–J99) | 11 | 4 | | 15 | 13.8 |
| Diseases of the digestive system (K00–K95) | 9 | 4 | | 13 | 11.9 |
| Multiple categories | 3 | 7 | | 10 | 9.2 |
| Endocrine, nutritional and metabolic diseases (E00–E89) | 4 | 3 | 2 | 9 | 8.3 |
| Injury, poisoning and certain other consequences of external causes (S00–T88) | 6 | 1 | | 7 | 6.4 |
| Neoplasms (C00–D49) | 5 | 1 | | 6 | 5.5 |
| Diseases of the musculoskeletal system and connective tissue (M00–M99) | 4 | 1 | | 5 | 4.6 |
| Diseases of the nervous system (G00–G99) | 2 | 3 | | 5 | 4.6 |
| Pregnancy, childbirth and the puerperium (O00–O9A) | 1 | 3 | | 4 | 3.7 |
| Certain infectious and parasitic diseases (A00–B99) | | 2 | | 2 | 1.8 |
| Diseases of the skin and subcutaneous tissue (L00–L99) | | 2 | | 2 | 1.8 |
| Diseases of the blood and blood-forming organs and certain disorders involving the immune mechanism (D50–D89) | | 1 | | 1 | 0.9 |
| Diseases of the genitourinary system (N00–N99) | 1 | | | 1 | 0.9 |
| Health status, including morbidity and/or mortality | 1 | | | 1 | 0.9 |
| Mental, behavioural and neurodevelopmental disorders (F01–F99) | | 1 | | 1 | 0.9 |
| N/A | | 8 | | 8 | 7.3 |

*Other study type category included: case definition validation by chart review (one study) and prescribing data quality assessment (one study).

†Six studies included multiple designs and, therefore, included each relevant category: case–control and cohort [three studies]; case-crossover and self-controlled case series (SCCS) (two studies) and cohort study and SCCS (one study).

ATC, Anatomical Therapeutic Chemical Classification; EHR, electronic health record; ICD-10, International Classification of Diseases 10th Revision.

studies, adjustment for clustering or effect heterogeneity was either not explored or not reported.

Whether and how to combine data is a key consideration in multi-database studies, and our results are consistent with a previous systematic review which found that 16 out of 22 multi-database pharmacoepidemiology studies used a one-stage approach to combine the data.[4]

One-stage meta-analysis approaches have gained popularity over the past two decades as a technique for combining individual participant' data from randomised controlled trials and other clinical studies that collect primary data, identified in systematic reviews.[30–32] One-stage meta-analysis has a number of advantages relevant to multi-database studies, which combine IPD from secondary data sources.[33] First, it ensures standardisation of the statistical analysis across all data sources. Second, it provides maximum flexibility to explore dose–response patterns, subgroup analyses and effect modification, all of which may help to account for heterogeneity between data sources. Third, a one-stage approach can incorporate information from smaller databases with sparse data, even where the database-specific effect cannot be reliably estimated due to zero cell counts.[28 34 35] However, one-stage meta-analysis of IPD should properly account for the clustered nature of the data from contributing databases,[27 36–39] since not doing so may introduce bias, especially if there is between-study heterogeneity in effect estimates. The results of this review suggest that clustering

**Table 2** Methodological aspects of analytical studies (N=62)

| Characteristic | Study design* | | | | | |
| --- | --- | --- | --- | --- | --- | --- |
| | Case–control studies | Cohort studies | Self-controlled studies | Other† | All | % |
| All studies | 23 | 34 | 7 | 2 | 62 | |
| Statistical methods‡ | | | | | | |
| Logistic regression | 23 | 8 | 2 | 1 | 34 | 54.8 |
| Poisson regression | | 6 | 6 | | 12 | 19.4 |
| Cox regression | | 18 | | | 18 | 29.0 |
| Other§ | | 9 | 1 | 1 | 11 | 17.7 |
| Confounder control‡ | | | | | | |
| Multiple regression or Mantel Haenszel test | 23 | 32 | | 2 | 55 | 88.7 |
| Matching | 23 | 9 | | 1 | 29 | 46.8 |
| Case only/self-controlled design | | | 7 | | 7 | 11.3 |
| Propensity scores | | 3 | | | 3 | 4.8 |
| Instrumental variables | | 2 | | | 2 | 3.2 |
| None | | 1 | | | 1 | 1.6 |
| Database comparisons/heterogeneity assessment‡ | | | | | | |
| Participant characteristics presented for each database | 17 | 24 | 4 | 2 | 45 | 72.6 |
| Effect estimates presented for each database | 18 | 19 | 5 | 2 | 41 | 66.1 |
| Formal test of effect heterogeneity | 10 | 4 | 3 | | 17 | 27.4 |
| $I^2$ | 6 | 3 | 3 | | 12 | 19.4 |
| Cochran's Q | 2 | 1 | | | 3 | 4.8 |
| Other or not specified | 3 | | | | 3 | 4.8 |
| No database comparisons (combined effect estimates only) | 5 | 11 | 2 | | 17 | 27.4 |
| Method for combining data or results‡ | | | | | | |
| Data not combined | 3 | 16 | 3 | 2 | 22 | 35.5 |
| Meta-analysis (two-stage) | 15 | 4 | 3 | | 21 | 33.9 |
| Random effects | 10 | 2 | 2 | | 13 | 21.0 |
| Fixed effects | 7 | 3 | 2 | | 13 | 21.0 |
| Method not specified | | 1 | | | 1 | 1.6 |
| Pooled analysis (one-stage) | 12 | 15 | 3 | | 29 | 46.8 |
| Multiple: one-stage and two-stage | 7 | 1 | 2 | | 10 | 16.1 |

*Six studies contributed to multiple categories because they included multiple designs: case–control and cohort (three studies); case-crossover and self-controlled case series (SCCS) (two studies) and cohort study and SCCS (one study).
†(One cross-sectional and one interrupted time series.
‡A single study could be included in more than one category.
§Other statistical methods included: negative binomial regression (two studies); Mantel-Haenszel test (two studies); two-stage instrumental variable (IV) models (two studies); 'data-mining methods' (two studies); generalised linear models (one study) and univariate tests (one study).

is largely ignored in multi-database studies using primary care EHR data, and this is consistent with findings from other reviews of one-stage meta-analysis in systematic reviews,[30 31] and in multi-database pharmacoepidemiology studies.[4 27] Barriers to the adoption of methods that properly account for clustering may include the perceived statistical complexity, lack of options in commonly available statistical software or because they can be computationally intensive.[27 40]

A third of analytical studies in this review used a two-stage approach to combine database-specific effect estimates to produce a pooled estimate. This approach avoids the need to share potentially sensitive IPD and may, therefore, be the only available option in some instances. It can also take advantage of local expertise and knowledge of each database partner, including optimising the use of available covariate information to control for confounding. In addition, a two-stage meta-analysis is relatively straightforward

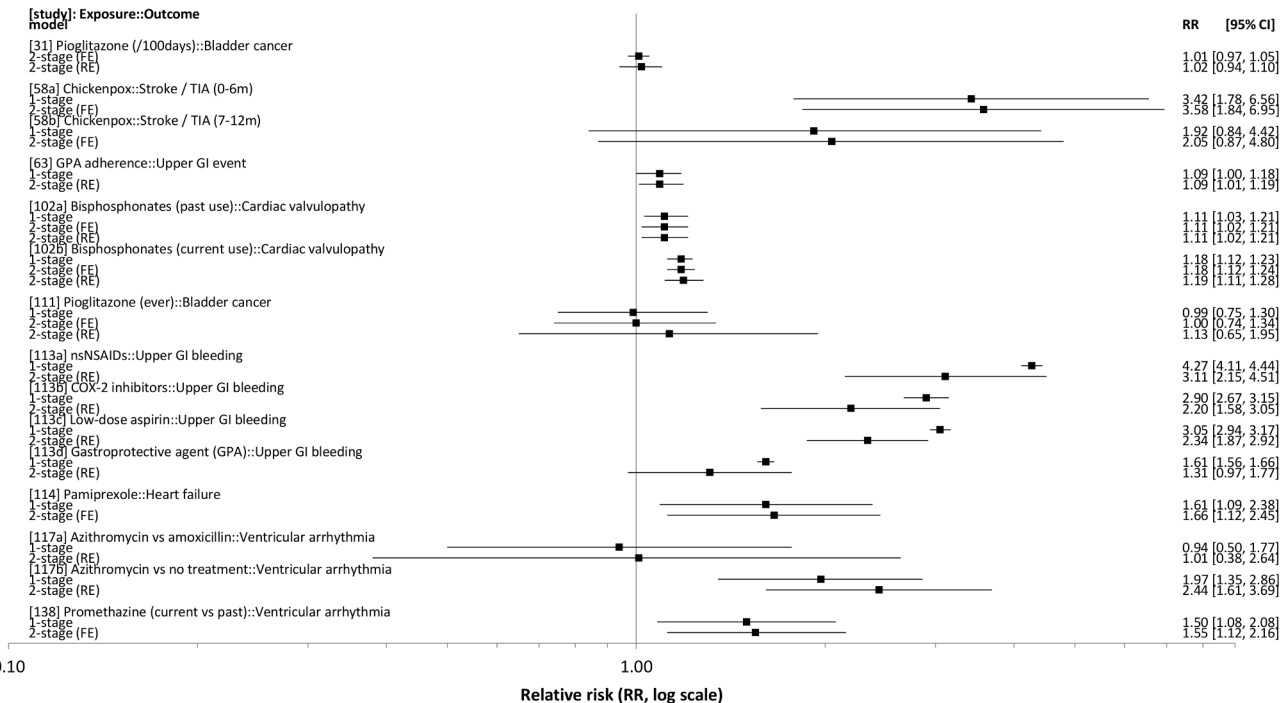

**Figure 2** Comparison of relative risk (RR) estimates reported in studies using two or more methods to combine data from multiple sources. [study] refers to the review database id number (see online supplemental table S1).

to implement and interpret. It is also possible to conduct pre-planned subgroup analyses, or examine dose–response effects using a two-stage approach, although sparse data may limit this, since databases with zero cell counts in one or both comparison groups will usually have to be excluded, unless some form of continuity correction is used, which can introduce bias.[41]

As for meta-analysis of randomised and prospective studies, a second key consideration for multi-database studies is how to assess and interpret results in the presence of heterogeneity. Limitations associated with using secondary data not collected for the specific study question impose an additional challenge in this context.[1][2] Several approaches for harmonising analyses of multi-database studies have been used. These include development and adoption of consistent and validated case definitions, use of common protocols and statistical analysis plans, and shared data management and analysis routines, all aiming to reduce external sources of variability in results from different databases.[10][11][16][17][19] Nevertheless, in studies incorporating data from different countries with different population characteristics and healthcare systems, these factors may contribute to real differences in the effect estimates. Even within a single country, different primary care EHR software systems are used, which may introduce heterogeneity in the extracted data.[42–47] With one-stage analysis, both clustering and heterogeneity can be naturally explored in hierarchical models incorporating FE and/or RE, though as already noted the tendency has been to ignore this issue in multi-database studies. For two-stage meta-analysis, we found that FE and RE models were used equally frequently and

model choice, where discussed at all, was generally related to the presence or absence of heterogeneity. Relatively few studies justified the choice based on some formal measure or test for heterogeneity, a practice which has been criticised because such tests often have low power. This is a problem especially when the number of studies or databases is small,[48] and can result in an FE model being used even though heterogeneity is present and an RE model may have been more appropriate.[49] Simulation studies have shown that the $I^2$ statistic can also be unreliable, either underestimating or overestimating heterogeneity in certain circumstances, particularly when the number of studies or databases is small.[50][51]

Several countries (Italy, Netherlands, Spain and the UK) now have two or more primary care EHR databases, and combining sources from the same country may reduce heterogeneity. In such cases, an FE model may be appropriate, especially where supporting analyses demonstrate substantial similarities in patient characteristics. Of eight single-country studies identified in this review and using two-stage meta-analysis, all eight used FE models[52–59]—despite evidence of substantial heterogeneity in some cases, although two studies did also use RE models for some analyses.[52][53]

When combining primary care EHR data from different countries or settings, an RE model might seem most appropriate, since these incorporate uncertainty in effect size when heterogeneity is present, yet reduce to an FE model if there is no heterogeneity.[48] However, when the number of estimates being combined is very small (<5) and heterogeneity is present—a common scenario in multi-database studies—conventional RE models may

**Table 3** Recommendations

| Recommendation | Rationale |
|---|---|
| Studies should report clearly on all aspects of study design and conduct which impact on harmonisation of analyses across data sources. | Allows assessment of the relative importance of heterogeneity induced by data management and analysis decisions vs heterogeneity inherent in the data. |
| Participant characteristics and effect estimates (where applicable) should be reported for each data source. | Assessment of heterogeneity is essential for interpretation, but formal methods for quantifying heterogeneity are inefficient and possibly biased in multi-database settings. |
| Where one-stage methods are used, studies should report whether and how analyses accounted for clustering and between database heterogeneity. | Interpretation requires understanding of extent to which heterogeneity might influence study results. |
| Where two-stage meta-analysis is used, studies should provide a clear rationale for choice of fixed effect (FE), random effects (RE) or other model. | Interpretation requires understanding of extent to which heterogeneity might influence study results. |
| Sensitivity analyses should include alternative methods for combining data. | Comparing the results of one-stage vs two-stage analyses, or FE vs RE models, provides information about potential impact of modelling assumptions. |
| Further research is needed to compare performance of one-stage and two-stage approaches for multi-database studies. | Relatively few studies have specifically addressed meta-analysis for multi-database studies. |

perform poorly. Simulation studies show that they can produce CIs which are too narrow, thereby increasing type one error rates.[60–63] A number of alternatives to conventional RE models have been proposed which partially address these limitations in some circumstances[60 63–66]; nevertheless, several authors have urged caution when interpreting results from meta-analysis of very few heterogeneous studies.[60 63 66] For multi-database studies, it may, therefore, be helpful to present estimates for both FE and RE models—or other alternative models, but always along with the results from individual databases.

Despite the differences outlined above, where studies combined data using more than one method, they produced similar estimates in most cases. However, in at least one study, one-stage and two-stage methods yielded large differences in both the point estimates and their precision. This may in part be related to the substantial heterogeneity in the database-specific estimates (reported $I^2$ between 86% and 98% for the estimates shown), but incomplete reporting of statistical methods limits further interpretation of these results. When the same modelling assumptions are used, one-stage and two-stage approaches are expected to give very similar results if the number of studies combined is relatively large.[37] However, few studies have systematically compared performance of one-stage and two-stage approaches for multi-database studies.[27 67]

### Limitations

The search strategy for this review included a list of named primary care EHR databases compiled from publicly available registers. This was to circumvent the poor sensitivity and specificity of conceptual searches based on MeSH terms, as reported in a previous review,[4] and confirmed in the current review. Our approach could have missed some eligible studies—if the abstract only mentioned primary care EHR data sources that were not in our list or did not mention the use of health databases or related terms at all. We would also have missed non-English language and abstract-only publications. Nevertheless, we expect the number of missed studies to be small, and any such studies are unlikely to have differed systematically from the included studies in terms of key methodological aspects. A more recent inventory of EHR databases in Europe did not identify any additional primary care databases that were not included in our review.[68] A further limitation was that some subjective interpretation was occasionally necessary to classify aspects of certain studies—for example, the rationale for combining, or the methods for managing and analysing data. The use of two reviewers helped to achieve some consistency across the included studies.

In conclusion, we found a growing body of literature reporting on studies using two or more sources of primary care EHR data. These addressed a range of research questions, and in many cases the results were presented separately and not combined. When data was combined, a one-stage meta-analysis was preferred. One-stage methods offer advantages in terms of analytical flexibility but are only possible where data management and governance arrangements allow for sharing of IPD. However, in many studies using one-stage approaches, the clustered nature of data from multi-database studies was frequently ignored, with unknown impact for interpretation. Two-stage meta-analysis requires only sharing of aggregated results, but there are known limitations with current two-stage methods when the number of studies is small, especially when some heterogeneity is expected. Irrespective of whether a one-stage or two-stage approach is used, combined results should be accompanied by results from each data source separately. This information, together with clear and complete reporting on methods used to

standardise and analyse the data (including the rationale for these decisions), affords a more considered assessment of potential sources of heterogeneity and greatly aids interpretation of the overall results. These considerations are relevant more widely to multi-database studies irrespective of the types of EHR or administrative databases being used, particularly where the number of databases being combined is relatively small, as was generally the case in our sample. Further research is required to understand the impact of analysis methods and other design aspects on overall study quality, and the development of reporting guidelines for multi-database studies, or extension of the existing RECORD guidance,[69] might be an important first step. Table 3 summarises key recommendations arising from our review.

**Acknowledgements** We would like to thank Russell Burke (Information Services, London School of Hygiene and Tropical Medicine) for his advice on bibliographic search strategies.

**Contributors** I confirm that all authors made substantial contributions to the study (detailed below), agree with and give final approval of the content of the current version, and agree to be accountable for all aspects of the work and for resolving questions related to any part of the work if they arise. DD: study concept and design, including search strategy; data extraction proformas and study database design; screening of titles and abstracts, and data extraction; analysis; interpretation of results; and drafting of manuscript. MC: screening of titles and abstracts, and data extraction; interpretation of results; critical review of manuscript and approval of final version. RW, KB and ID: study concept and design, including search strategy; interpretation of results; critical review of manuscript and approval of final version. SE: interpretation of results, critical review of manuscript and approval of final version.

**Funding** This research was conducted as part of a postgraduate doctoral degree funded by the Clinical Practice Research Datalink. KB holds a Sir Henry Dale Fellowship funded by Wellcome and the Royal Society (grant number 107731/Z/15/Z).

**Competing interests** None declared.

**Patient and public involvement** Patients and/or the public were not involved in the design, or conduct, or reporting, or dissemination plans of this research.

**Patient consent for publication** Not required.

**Provenance and peer review** Not commissioned; externally peer reviewed.

**Data availability statement** Data are available upon reasonable request. The study database and data-extraction proformas are available on request from DD.

**ORCID iDs**
Daniel Dedman http://orcid.org/0000-0002-3699-5391
Melissa Cabecinha http://orcid.org/0000-0001-6869-4692

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
