## [Reviewer comments · BMJ Open]

ARTICLE DETAILS

TITLE (PROVISIONAL)	Approaches for combining primary care electronic health record data from multiple sources: a systematic review of observational studies
AUTHORS	Dedman, Daniel; Cabecinha, Melissa; Williams, Rachael; Evans, Stephen; Bhaskaran, Krishnan; Douglas, Ian

VERSION 1 – REVIEW

REVIEWER	Persephone Doupi THL, Finland
REVIEW RETURNED	04-Mar-2020

GENERAL COMMENTS	1. In the Methods section the exclusion criteria are listed as follows: “It excluded data generated primarily for administrative purposes such as health insurance claims data, where the motivation for recording is different. Apart from the specific focus on primary care EHR databases, no other restrictions were applied in terms of populations, geography, study period, exposure, outcome or study design.” However, in the Review Protocol (Appendix S1, Rational and scope section) it is stated that: “The review specifically covered 'horizontal' combination of data from different sources, containing data from different sets of individuals (possibly after deduplication). Here the primary purpose of combining might be to increase the number of individuals available for analysis, or increase the range of population settings to which findings can be applied (i.e. increase external validity). The review did not include studies where data sources were only combined 'vertically' i.e. linkage studies whereby data on the same individuals was combined to provide richer (deeper) information about each study participant.” Combination of data sources in order to acquire access to all variables relevant for a specific research is a very common motive of studies using multiple data sources. Presumably, the exclusion of such linkage studies took primarily place at the screening phase (since there is only mention of one linkage study excluded at the full-text phase). "Vertical" linkage studies should therefore also be clearly listed as an exclusion criterion, accompanied with the logic for its use – particularly since “The focus of the review was on describing the range of multi-database studies and methods for combining primary care EHR data” (shouldn't that range then also have included vertical linkage studies?). Moreover, since the team included studies which used also other data sources in addition to at least two primary care EHR databases, it is uncertain whether at least some vertical linkage studies were included after all.
--

	2. The authors use the abbreviation IPD for “individual patient data”. Although the abbreviation is, of course, correct, it propagates the misleading perception that “individual patient data” is an interchangeable term to “individual participant data”, which is also (correctly) abbreviated to ‘IPD’. Meta-analysis of individual participant data, the methodological evolution of the systematic review “classic” meta-analysis of published research data (e.g. Riley RD et al. BMJ 2010;340:c221 doi: 10.1136/bmj.c221), has a longer history than “individual patient data” meta-analysis, which is a relatively recent development enabled by the digitalization of healthcare data. Most importantly though, there are significant differences in the quality of the data involved in the two approaches. As Hall et al. (Pharmacoepidemiology and drug safety 2012; 21: 1–10) have nicely summarized the challenge of secondary use of data (as the use of primary EHR databases is), “studies must be performed within the limitations of a resource not specifically designed to test the research hypothesis but the product of complex and evolving healthcare systems”. Also the process of identifying appropriate resources and acquiring access to data is very different between access to published study data sets vs. those residing in collections created through routine clinical documentation. I would therefore encourage the authors to review their text keeping this crucial distinction in mind and make the respective clarifications where necessary, e.g. in the discussion where references representing these two different research work trajectories are pooled together [30-33]. 3. Minor detail: Given the acknowledged difficulties in running bibliographic searches on multi-database studies and the resulting approach adopted by the authoring team (starting from specific compendiums of primary care databases) – limitations also reported by the authors – there is no need to say in the Article summary that this is a “full range of completed studies” (even if that was the aim at the outset of the work, and hence an acceptable statement in the Introduction); an “as comprehensive as possible coverage” of such studies would be closer to the truth.
--	---

REVIEWER	Jan-David Liebe Hochschule Osnabrück University of Applied Sciences
REVIEW RETURNED	24-Apr-2020

GENERAL COMMENTS	Basically, I have only a few remarks, since it is a well-structured and methodologically sound work with recognizable relevance. The motivation and objectives were clearly defined. The method is presented in a clear and comprehensible way. With the chosen search strategy, an echo chamber problem does not seem entirely unlikely, but the authors provide a reasonable explanation for the chosen procedure. Cross-review on a randomized 20% sample is common and useful, even if subjective judgements cannot be completely excluded. The selected categorizations appear to be appropriate. By conducting my own minimal sample, I was able to verify the results. Although the descriptive results do not seem surprising, they do raise further questions, which the authors, discuss extensively. For
--

	example, it is obvious that there are a number of different methods to reduce external sources of variability of results from different databases (the review illustrates this). At the same time, it is clear that differences in effect estimates can occur depending on the choice of method. The systematic review and presentation of these methodological diversity raises awareness of how carefully results must be evaluated and interpreted in the case of heterogeneity between databases. The authors derive practical advice and a guideline from their results on how to reduce heterogeneity and for the question in which cases FE or RE models should be used. As a next step, it would be helpful to know the extent to which the method identified for combining and analyzing EHR data is related to the quality of the study results.
--	--

REVIEWER	Griffin M Weber Harvard Medical School, United States
REVIEW RETURNED	11-May-2020

GENERAL COMMENTS	The authors reviewed 109 articles describing studies that used data from more than one primary care EHR. The types of information they report include: (1) the study objective (drug safety, disease risk, etc.); (2) rationale for using multiple data sources; (3) number of EHRs and methods for combining the data (merging individual patient records, combining locally calculated statistics, etc.); (4) study design (case-control, cohort, etc.); and (5) data management (distributed vs central); and others. The strength of this manuscript is that it highlights the many different ways that studies utilize data from multiple EHRs. This really emphasizes the need for studies to consider different options and provide a rationale for the one they use. The authors provide useful recommendations and point out the types of statistical analyses and reporting that studies should ideally do. I've seen other comparisons of individual components of this, such as distributed vs centralized data management. However, the authors provide a nice comprehensive list of study characteristics. The supplement data include details on each of the 109 articles, the search terms used to find these articles, and details of different methods they used to find articles. A few minor points: 1) The authors use the terms "partially aggregated" and "semi-aggregated". I'm not sure what these mean and whether they are the same or different. The second paragraph on page 12 has "a 1-stage meta-analysis of pooled individual patient data or partially aggregated data was undertaken in 29 studies". Is this stating that that partially aggregate data is a 1-stage meta-analysis? How is partially aggregated data different than "2-stage" or "fully distributed"? 2) The authors suggest that they have captured the majority of studies involving two or more primary care EHRs. It would be good to note that this might represent only a small fraction of all studies that use data from two or more EHRs (including primary care EHRs, non-primary care EHRs, and EHRs that contain both primary care and non-primary care data). Furthermore, they acknowledge most of their primary care EHRs are from Europe. This probably isn't a comprehensive list of all primary care practices around the world
---

	that have an EHR system. What they really did is collect the majority of studies involving two or more data sources from a particular list of (mostly European) primary care EHRs. This is important because the things that I think are strengths of this paper, such as their recommendations, are generalizable to any multi-EHR study. I think they should mention this as a broader potential impact of their analysis. At the same time, though, their numeric results, such as the percentages of studies that use different methods, could be very specific to their list of EHRs. One thing that stood out to me was that none of the 109 studies had more than 4 primary care EHRs and 8 total EHRs. They noted that none of the 109 studies used a global common data model (CDM, OMOP, Sentinel, etc.). I know that some studies that leverage these global networks link dozens of EHRs. It would be interesting as a future analysis to see how these larger networks fit into the authors framework. The authors might want to comment about that, and the challenges or benefits of combining data from a much larger number of EHRs. 3) I don't understand the part about "a second reviewer screened a 20% random sample of all abstracts". Was this to verify the results of the first reviewer, or to find additional papers? Did both reviewers have to determine the paper was eligible for inclusion, thus reducing the number of final papers by 80%
--	--

VERSION 1 – AUTHOR RESPONSE

Reviewer: 1

1. In the Methods section the exclusion criteria are listed as follows:

"It excluded data generated primarily for administrative purposes such as health insurance claims data, where the motivation for recording is different. Apart from the specific focus on primary care EHR databases, no other restrictions were applied in terms of populations, geography, study period, exposure, outcome or study design."

However, in the Review Protocol (Appendix S1, Rational and scope section) it is stated that:

"The review specifically covered 'horizontal' combination of data from different sources, containing data from different sets of individuals (possibly after deduplication). Here the primary purpose of combining might be to increase the number of individuals available for analysis, or increase the range of population settings to which findings can be applied (i.e. increase external validity). The review did not include studies where data sources were only combined 'vertically' i.e. linkage studies whereby data on the same individuals was combined to provide richer (deeper) information about each study participant."

Combination of data sources in order to acquire access to all variables relevant for a specific research is a very common motive of studies using multiple data sources. Presumably, the exclusion of such linkage studies took primarily place at the screening phase (since there is only mention of one linkage study excluded at the full-text phase).

"Vertical" linkage studies should therefore also be clearly listed as an exclusion criterion, accompanied with the logic for its use – particularly since "The focus of the review was on describing the range of multi-database studies and methods for combining primary care EHR data" (shouldn't that range then also have included vertical linkage studies?).

Moreover, since the team included studies which used also other data sources in addition to at least two primary care EHR databases, it is uncertain whether at least some vertical linkage studies were included after all.

Author's response:

Thank you for highlighting this potential ambiguity. We did not specifically exclude "vertical" linkage studies. A primary care EHR database could be included even if it was linked vertically to another database, but it was only counted as a single database

We have added the following text to clarify:

[Methods, paragraph 1]: Primary care EHR databases were considered irrespective of whether they were “vertically” linked (i.e. linked at the individual patient level) to another data source such as a disease registry or dispensing database. Each “vertically” linked primary care EHR database was treated as a single data source.

[end response]

2. The authors use the abbreviation IPD for “individual patient data”. Although the abbreviation is, of course, correct, it propagates the misleading perception that “individual patient data” is an interchangeable term to “individual participant data”, which is also (correctly) abbreviated to ‘IPD’. Meta-analysis of individual participant data, the methodological evolution of the systematic review “classic” meta-analysis of published research data (e.g. Riley RD et al. *BMJ* 2010;340:c221 doi: 10.1136/bmj.c221), has a longer history than “individual patient data” meta-analysis, which is a relatively recent development enabled by the digitalization of healthcare data. Most importantly though, there are significant differences in the quality of the data involved in the two approaches. As Hall et al. (*Pharmacoepidemiology and drug safety* 2012; 21: 1–10) have nicely summarized the challenge of secondary use of data (as the use of primary EHR databases is), “studies must be performed within the limitations of a resource not specifically designed to test the research hypothesis but the product of complex and evolving healthcare systems”.

Also the process of identifying appropriate resources and acquiring access to data is very different between access to published study data sets vs. those residing in collections created through routine clinical documentation. I would therefore encourage the authors to review their text keeping this crucial distinction in mind and make the respective clarifications where necessary, e.g. in the discussion where references representing these two different research work trajectories are pooled together [30-33].

Author’s response:

Thank you – we agree with both your points. We have removed the IPD acronym throughout the main text, writing the relevant term out as appropriate, and added text to highlight the distinction between systematic reviews combining primary data collected on individual participants, and multi-database studies using secondary data from individual patients.

Added text:

[Discussion, paragraph 2]: One-stage meta-analysis approaches have gained popularity over the past 2 decades as a technique for combining individual participant data from randomised controlled trials and other clinical studies that collect primary data, identified in systematic reviews.[30–32] One-stage meta-analysis has a number of advantages relevant to multi-database studies which combine individual patient data from secondary data sources.[33]

[Discussion, paragraph 4]: As for meta-analysis of randomised and prospective studies, a second key consideration for multi-database studies is how to assess and interpret results in the presence of heterogeneity. Limitations associated with using secondary data not collected for the specific study question impose an additional challenge in this context.[1,2]

[end response]

3. Minor detail: Given the acknowledged difficulties in running bibliographic searches on multi-database studies and the resulting approach adopted by the authoring team (starting from specific compendiums of primary care databases) – limitations also reported by the authors – there is no need to say in the Article summary that this is a “full range of completed studies” (even if that was the aim at the outset of the work, and hence an acceptable statement in the Introduction); an “as comprehensive as possible coverage” of such studies would be closer to the truth.

Author’s response:

We agree and have amended the statement in the Article summary as suggested [Bullet point 3].

[end response]

Reviewer: 2

Basically, I have only a few remarks, since it is a well-structured and methodologically sound work with recognizable relevance.

The motivation and objectives were clearly defined. The method is presented in a clear and comprehensible way.

With the chosen search strategy, an echo chamber problem does not seem entirely unlikely, but the

authors provide a reasonable explanation for the chosen procedure.

Cross-review on a randomized 20% sample is common and useful, even if subjective judgements cannot be completely excluded. The selected categorizations appear to be appropriate. By conducting my own minimal sample, I was able to verify the results.

Although the descriptive results do not seem surprising, they do raise further questions, which the authors, discuss extensively. For example, it is obvious that there are a number of different methods to reduce external sources of variability of results from different databases (the review illustrates this). At the same time, it is clear that differences in effect estimates can occur depending on the choice of method. The systematic review and presentation of these methodological diversity raises awareness of how carefully results must be evaluated and interpreted in the case of heterogeneity between databases.

The authors derive practical advice and a guideline from their results on how to reduce heterogeneity and for the question in which cases FE or RE models should be used. As a next step, it would be helpful to know the extent to which the method identified for combining and analyzing EHR data is related to the quality of the study results.

Author's response:

Thank you for this. We agree on the need for more research to understand determinants of study quality including analytical methods, and have amended the discussion to note this.

Added text:

[Discussion, last paragraph] Further research is required to understand the impact of analysis methods and other design aspects on overall study quality, and the development of reporting guidelines for multi-database studies, or extension of the existing RECORD guidance,[70] might be an important first step.

[end response]

Reviewer: 3

The authors reviewed 109 articles describing studies that used data from more than one primary care EHR. The types of information they report include: (1) the study objective (drug safety, disease risk, etc.); (2) rationale for using multiple data sources; (3) number of EHRs and methods for combining the data (merging individual patient records, combining locally calculated statistics, etc.); (4) study design (case-control, cohort, etc.); and (5) data management (distributed vs central); and others.

The strength of this manuscript is that it highlights the many different ways that studies utilize data from multiple EHRs. This really emphasizes the need for studies to consider different options and provide a rationale for the one they use. The authors provide useful recommendations and point out the types of statistical analyses and reporting that studies should ideally do. I've seen other comparisons of individual components of this, such as distributed vs centralized data management. However, the authors provide a nice comprehensive list of study characteristics. The supplement data include details on each of the 109 articles, the search terms used to find these articles, and details of different methods they used to find articles.

A few minor points:

1) The authors use the terms "partially aggregated" and "semi-aggregated". I'm not sure what these mean and whether they are the same or different. The second paragraph on page 12 has "a 1-stage meta-analysis of pooled individual patient data or partially aggregated data was undertaken in 29 studies". Is this stating that that partially aggregate data is a 1-stage meta-analysis? How is partially aggregated data different than "2-stage" or "fully distributed"?

Author's response:

Thank you for pointing out this inconsistency. We used "semi-aggregated" throughout the original review protocol (appendix), but came to prefer "partially aggregated" by the time of writing the manuscript. We consider the terms interchangeable, and have added text for clarification in the methods section:

[Methods, paragraph 5]: Partially (or semi-) aggregated data summarises information on more than one individual (thereby enhancing privacy protection) while still allowing the pooling of data across databases for further analysis, including 1-stage meta-analysis. Examples include total person time and event counts for groups of patients sharing the same characteristics.

[end response]

2) The authors suggest that they have captured the majority of studies involving two or more primary

care EHRs. It would be good to note that this might represent only a small fraction of all studies that use data from two or more EHRs (including primary care EHRs, non-primary care EHRs, and EHRs that contain both primary care and non-primary care data). Furthermore, they acknowledge most of their primary care EHRs are from Europe. This probably isn't a comprehensive list of all primary care practices around the world that have an EHR system. What they really did is collect the majority of studies involving two or more data sources from a particular list of (mostly European) primary care EHRs. This is important because the things that I think are strengths of this paper, such as their recommendations, are generalizable to any multi-EHR study. I think they should mention this as a broader potential impact of their analysis. At the same time, though, their numeric results, such as the percentages of studies that use different methods, could be very specific to their list of EHRs. One thing that stood out to me was that none of the 109 studies had more than 4 primary care EHRs and 8 total EHRs. They noted that none of the 109 studies used a global common data model (CDM, OMOP, Sentinel, etc.). I know that some studies that leverage these global networks link dozens of EHRs. It would be interesting as a future analysis to see how these larger networks fit into the authors framework. The authors might want to comment about that, and the challenges or benefits of combining data from a much larger number of EHRs.

Author's response:

Thank you – we are pleased the broader relevance of this review is apparent and have tried to make this more explicit, while acknowledging that our sample did not include studies using very many databases. We agree that further exploration of this issue would be of value, but would have extended the scope beyond the sample at hand, and would have been difficult to address fully in the space available.

We have added the following text:

[Discussion, last paragraph]: These considerations are relevant more widely to multi-database studies irrespective of the types of EHR or administrative databases being used, particularly where the number of databases being combined is relatively small, as was generally the case in our sample.

[end response]

3) I don't understand the part about "a second reviewer screened a 20% random sample of all abstracts". Was this to verify the results of the first reviewer, or to find additional papers? Did both reviewers have to determine the paper was eligible for inclusion, thus reducing the number of final papers by 80%

Author's response:

The purpose of the second reviewer screening a random 20% of abstracts was to check that application of inclusion and exclusion criteria and thus selection of studies into the final list was reliable. No additional studies were selected by the second reviewer during this process and we were therefore satisfied that screening by a single reviewer was sufficient.

We have amended the text to clarify:

[Methods, paragraph 3]: Titles and abstracts of all retrieved studies were screened for eligibility by one reviewer (DD). A random 20% sample was also screened by a second reviewer (MC) and showed very good agreement between the two.

[end response]

VERSION 2 – REVIEW

REVIEWER	Griffin M Weber Harvard Medical School, United States of America
REVIEW RETURNED	27-Jun-2020
GENERAL COMMENTS	The authors clarified the questions I previously had about partially- vs semi- aggregated data and the purpose of the second reviewer. I see that in response to another reviewer, the authors changed "full and increasing range of completed studies" down to "as comprehensive coverage as possible". I still think they are only

	looking at a small biased sample of studies that used two or more EHRs. However, that does not affect the overall message of this manuscript. So, “as comprehensive coverage as possible” is fine.
--	--